Corrected: Author correction

# Hybridization is a recurrent evolutionary stimulus in wild yeast speciation

Chris Eberlein[1,2,3,4], Mathieu Hénault[1,3,4,5], Anna Fijarczyk[1,2,3,4], Guillaume Charron [1,2,3,4], Matteo Bouvier[1,2,3], Linda M. Kohn[6], James B. Anderson [6] & Christian R. Landry [1,2,3,4,5]

Hybridization can result in reproductively isolated and phenotypically distinct lineages that evolve as independent hybrid species. How frequently hybridization leads to speciation remains largely unknown. Here we examine the potential recurrence of hybrid speciation in the wild yeast *Saccharomyces paradoxus* in North America, which comprises two endemic lineages *SpB* and *SpC*, and an incipient hybrid species, *SpC**. Using whole-genome sequences from more than 300 strains, we uncover the hybrid origin of another group, *SpD*, that emerged from hybridization between *SpC** and one of its parental species, the widespread *SpB*. We show that *SpD* has the potential to evolve as a novel hybrid species, because it displays phenotypic novelties that include an intermediate transcriptome profile, and partial reproductive isolation with its most abundant sympatric parental species, *SpB*. Our findings show that repetitive cycles of divergence and hybridization quickly generate diversity and reproductive isolation, providing the raw material for speciation by hybridization.

[1] PROTEO, The Quebec Network for Research on Protein Function, Engineering, and Applications, Québec, QC G1V 0A6, Canada. [2] Département de Biologie, Université Laval, Québec, QC G1V 0A6, Canada. [3] Institut de Biologie Intégrative et des Systèmes (IBIS), Université Laval, 1030 Ave de la Médecine, Québec, QC G1V 0A6, Canada. [4] Centre de recherche en données massives (CRDM), Université Laval, Québec, QC G1V 0A6, Canada. [5] Département de Biochimie, Microbiologie et Bio-informatique, Université Laval, Québec, QC G1V 0A6, Canada. [6] Departments of Ecology and Evolutionary Biology and Cell and Systems Biology, University of Toronto Mississauga, 3359 Mississauga Rd, Mississauga, ON L5L 1C6, Canada. These authors contributed equally: Chris Eberlein, Mathieu Hénault. Correspondence and requests for materials should be addressed to C.E. (email: Chris.Eberlein.1@ulaval.ca) or to C.R.L. (email: Christian.Landry@bio.ulaval.ca)

Hybridization among species is considered as a stimulating force in evolution[1], but its consequences are difficult to predict. On one hand, it can lead to the collapse of species barriers[2], and on the other, to the formation of new species[3]. Hybrids can benefit from extreme phenotypes that allow them to exploit novel ecological niches[4] or simply new combinations of traits that offer a strong advantage over the parents[5]. The frequency at which species arise from hybridization and under which circumstances this process takes place remain to be fully understood[6,7].

Hybridization among eukaryotic microbes has been shown to be a powerful mean to create genomic and phenotypic diversity[8,9], however its contribution to the evolution of natural populations still has to be fully investigated. Studies of fungi in close association with humans suggest that hybridization could be a key driver of microbial diversity. For instance, animal and plant fungal pathogens[10] rely on hybridization to colonize new hosts and acquire virulence traits. In addition, budding yeasts of the genus *Saccharomyces* have drawn diversity from interspecific crosses, with tens of domesticated strains that adapted to industrial environments at least partly, thanks to their hybrid origin[11]. Population genomics studies have also shown that interspecies introgression could contribute to yeast genomic diversity in nature[12], although these studies involve at least one partially domesticated species in most instances. Documenting the evolutionary and ecological conditions in which hybridization events take place and their consequences in completely natural systems is imperative because most of the microbial diversity present today has evolved in the absence of human influence.

A recent population genomics study of *Saccharomyces paradoxus*, a budding yeast found worldwide on the bark of deciduous trees and their associated soils[13,14], showed that a novel North American species evolved through hybridization about 10,000 years ago[15]. This hybrid species (*SpC\**) originated from the secondary contact between the two most abundant species, *SpB* that occupies a large fraction of the continent, and *SpC*[15], which is found almost exclusively so far in the north east. *SpC\** shows a unique profile of growth phenotypes[15], occurs mostly in the zone of sympatry between its two parental species and shows reproductive isolation with both of them, which is caused at least partially by genome rearrangements. These findings revealed that hybridization occurred between two incipient species (*SpB* and *SpC*) that originated a little more than 100,000 years ago and that it led to the formation of *SpC\**. A recent study by Xia et al.[16] identified a novel group of strains, *SpD* (originally defined as "Clade d" and then mistakenly assigned to the *SpC\** group), which exhibit signatures of genomic admixture, potentially involving the same parental species as *SpC\**. Analyses by Hénault et al.[17] suggested that *SpD* could have arisen from a second hybridization between *SpB* and *SpC*, indicating that hybridization could have occurred multiple times in different locations[16,17].

With additional sampling, genome sequencing and systematic phenotyping, we find that *SpD* strains are recent hybrids between the hybrid species *SpC\** and the most abundant North American lineage, *SpB*. This backcross between a hybrid and one of its parental species generated novel growth phenotypes, intermediate transcriptional profiles, a novel genome architecture and partial reproductive isolation. These results highlight that speciation in yeast may result from repeated cycles of divergence and hybridization.

## Results

**A new hybrid lineage in North America**. We examined whether the origin of *SpD* can be traced back to a hybridization event between the previously described endemic *S. paradoxus* species *SpB* and *SpC*. We assessed the population structure and genetic relationship from fully sequenced genomes of 316 *S. paradoxus* strains, which included 38 newly sampled strains (2016), 34 strains previously sampled, 91 genomes from Xia et al.[16], and 153 genomes from Leducq et al.[15] (Fig. 1, Supplementary Figure 1, Supplementary Data 1-2). In agreement with previous phylogenetic analyses[16,17], we found that the strains form five clusters corresponding to *SpA*, *SpB*, *SpC*, *SpC\**, and *SpD* (Fig. 1b, c, Supplementary Figure 2). The two main lineages *SpB* and *SpC* exhibit nucleotide divergence of 2.2% on average (Supplementary Figure 3). *SpB* shows high nucleotide diversity (0.42%), which likely results from its large geographic distribution and subpopulation structure (Supplementary Figure 4). *SpA* exhibits little diversity (0.1%), which is consistent with its recent introduction from Europe[15]. The *SpD* group lies in between the two main lineages *SpB* and *SpC*, next to the hybrid species *SpC\** and exhibits the highest level of within-group nucleotide diversity (0.6%) (Supplementary Figure 3).

We used population genetics analysis (*f4* statistics) to test different hypotheses about admixture. Detecting admixture between lineages would explain their evolutionary relationships and, if so, decipher the *SpD* ancestry relative to *SpB*, *SpC*, and *SpC\**. The genomic composition of *SpD* is best supported by two models involving admixture: either *SpD* arose from admixture between *SpB* and the hybrid species *SpC\** (M01) or between *SpB* and *SpC* before the origin of *SpC\** (M02; Fig. 1d, e, Supplementary Figures 5-9). These results confirm that *SpD* is a second hybrid lineage related to *SpC\**, but they do not allow us to disentangle which of *SpC\** or *SpD* emerged first.

**Hybridization between a hybrid species and its parent**. The geographical distributions of North American endemic *S. paradoxus* lineages support the first of the two admixture models (M01), suggesting that *SpD* originated from hybridization between *SpB* and *SpC\**. All of the *SpD* strains were isolated from a small area near Toronto, Canada[16], which is part of the wide distribution of *SpB*. In contrast, neither *SpC* nor *SpC\** had been isolated from this area prior to the current study, in spite of intensive sampling efforts[16]. Our most recent sampling and genotyping of *S. paradoxus* isolates revealed a single *SpC\** strain isolated approximately 2 km away from *SpD* sampling site (Fig. 1a, Supplementary Data 1). This suggests that although rare, *SpC\** is present in this *SpB*-dominated area and makes natural hybridization scenarios between *SpB* and *SpC\** possible.

The recent descent of *SpD* from the hybridization of *SpB* with the hybrid species *SpC\** (M01) is supported by two largely independent data sets: (i) structural genomic variation and (ii) discrete introgressed regions in the hybrid genomes. We examined structural genomic variation by comparing six new de novo genome assemblies produced from Oxford Nanopore long reads (Supplementary Table 1) and one published assembly (YPS138, *SpB*) produced from PacBio long reads[18]. We confirmed that chromosomal inversions and translocations segregate among *SpA*, *SpB*, *SpC*, and *SpC\**[15,19] (Fig. 2b). We also found that some of these rearrangements segregate within *SpD*. The *SpD* lineage was previously suggested to comprise two subclades[17] and its within-lineage nucleotide diversity was the highest of all lineages, demonstrating the role of hybridization in generating genomic diversity (Supplementary Figure 3). We thus separated the *SpD* strains according to this classification, into two sub-groups named *SpD1* and *SpD2*. In the two most parsimonious tree topologies based on chromosomal inversions and translocations (Fig. 2a, b; Supplementary Figure 10), *SpD1* and *SpD2* formed a monophyletic group with *SpB* and *SpC\**, while *SpC* was more divergent. Inversion i3 on chromosome V and translocation t1

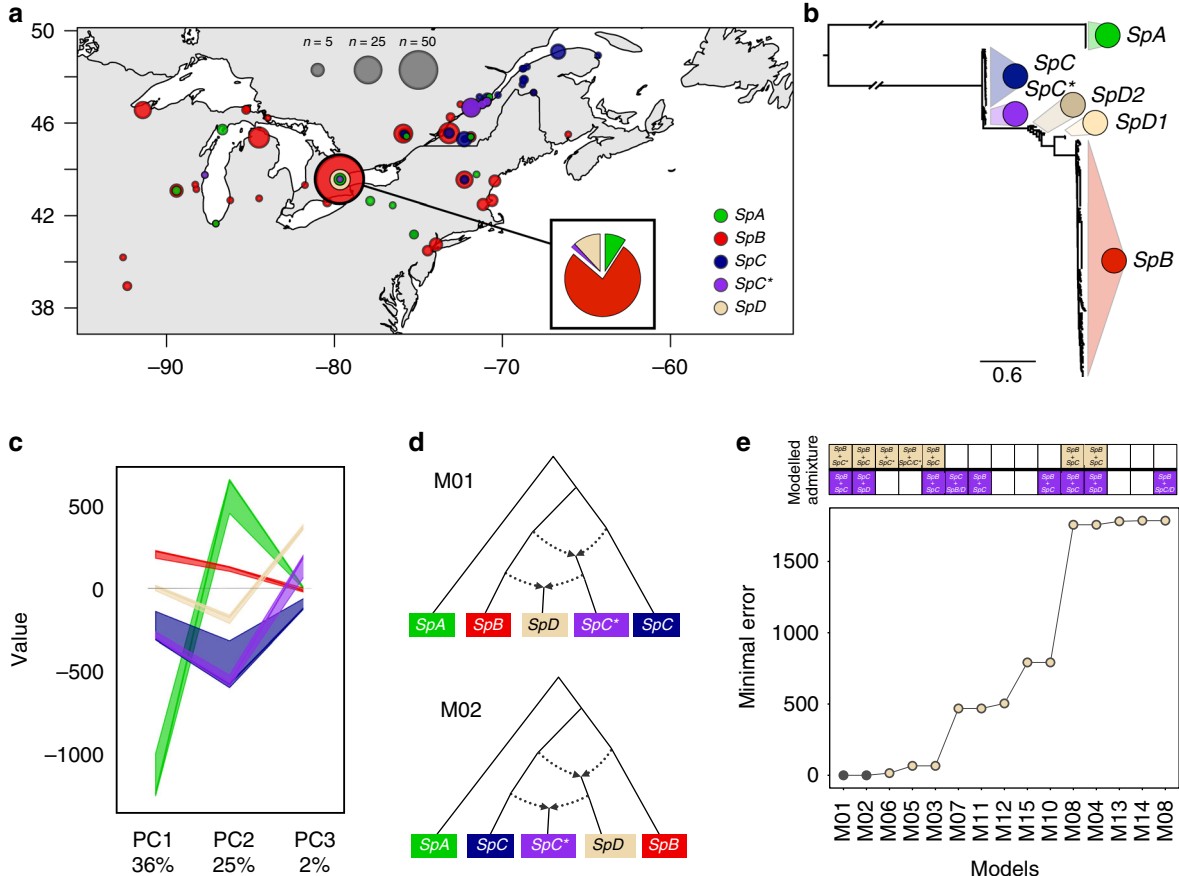

**Fig. 1** Population structure of *Saccharomyces paradoxus* in North America. **a** Sampling locations (circles) of 316 whole-genome sequenced strains from five distinct groups: *SpA*, *SpB*, *SpC*, *SpC\**, and *SpD*[15-17]. The map was drawn with R v3.3.3[82] using the package maps (version 3.3.0). Detailed strain information can be found in Supplementary Data 1. **b** Genetic relationship among isolates from a maximum likelihood phylogenetic tree built from 25,280 variable positions. *SpD* splits into the two sub-clades *SpD1* and *SpD2* as previously suggested by Hénault et al.[17]. **c** Grouping of strains by principal component analysis showing the relationship among lineages. PC1 to PC3 separate *SpA*, *SpB*, *SpC*, *SpC\**, and *SpD* as independent genetic clusters (For color code, see Fig. 1a). The spread of strains (median and the 20th to 80th percentile range) in each PC (x-axis) is shown. **d** The two models (M01, M02) from the *f4* statistics explain admixed ancestry of *SpC\** and *SpD*. Dotted arrows indicate admixture events. **e** Ranking of the 15 models based on their fit to the data. Filled squares on the top indicate if the models assume admixed ancestry of *SpD* and *SpC\**. The five best models indicate hybrid origin of *SpD*. The two best models matching all *f4* statistics (filled dots) suggest a hybrid origin for *SpD* resulting from a cross between *SpB* and the hybrid *SpC\** (model M01), or hybrid origin of *SpD* resulting from a cross between *SpB* and *SpC*, occurring before the origin of a *SpC\** hybrid (model M02)

between chromosomes VI and XIII (corresponding to VItXIII described in Leducq et al.[15]) are shared by *SpC\** and *SpD*, while inversion i5 on chromosome VI is shared by *SpB*, *SpC\**, and *SpD* (Fig. 2c). In addition, inversion i1 on chromosome X is common to *SpB* and *SpD2*. However, some rearrangements (e.g. i2) conflict with these most parsimonious topologies. These conflicts may originate from hybridization, although incomplete lineage sorting or independent occurrence due to inversion or translocation hotspots could explain them as well. Nevertheless, the most parsimonious scenarios clearly show that both *SpD1* and *SpD2* share more structural genomic similarity with *SpC\** than *SpC*, supporting its *SpB-SpC\** origin.

Having the strongest support for the model under which *SpD* originated from a cross between *SpB* and *SpC\**, we mapped the regions of genomic admixture of *SpD* strains to reference genomes from *SpB* and *SpC\**, which enabled to examine the patterns and size of parental genomic regions in the *SpD* strains (Fig. 3a). *SpD* strains have about 50% *SpB* and 50% *SpC\** of genomic origin (Figure 3Aii). The patterns of *SpB*- and *SpC\**-like regions allowed us to distinguish between the two *SpD* sub-groups, as seen from the genome-wide phylogenies[17]. Meiotic recombination and asymmetrical parental backcrossing affect the pattern of parental genomic

blocks in hybrid genomes. Through meiotic recombination, uniparental backcrossing enlarges the size of blocks that share ancestry with the backcrossing parent[20]. In the hybrid species *SpC\**, our analysis detects ~3% introgression from *SpB* (minor parent). The minor proportion of *SpB* elements shows that crossing-overs from meiotic recombination and backcrosses with *SpC* have reduced the genomic blocks of *SpB* ancestry to a few regions.

These introgressed regions could play an important role in the ecological divergence of *SpC\**, as fixed *SpB* introgressions are marginally enriched for genes involved in the response to amino acids[17] (Supplementary Figure 11, Supplementary Table 2–3). We compared the size of *SpB* parental introgression blocks between *SpC\** and *SpD*, as longer blocks in *SpD* could confirm its most recent origin. The length of *SpB*-like blocks in *SpD* is significantly greater than in *SpC\** (median of 12.2 kb and 1.1 kb respectively; Mann-Whitney *U*-test, *P*-value < 2.2e−16, Fig. 3b). Since the fragments of both *SpB* and *SpC\** origin represent about equal proportions in *SpD* genomes, it seems unlikely that *SpD* experienced extensive backcrossing. Indeed, the block size distributions of *SpB* and *SpC\** elements in *SpD* genomes were not significantly different (median of 12.2 kb and 10.4 kb, respectively; Mann-Whitney *U*-test, *P*-value 0.99). We thus hypothesize that *SpD*

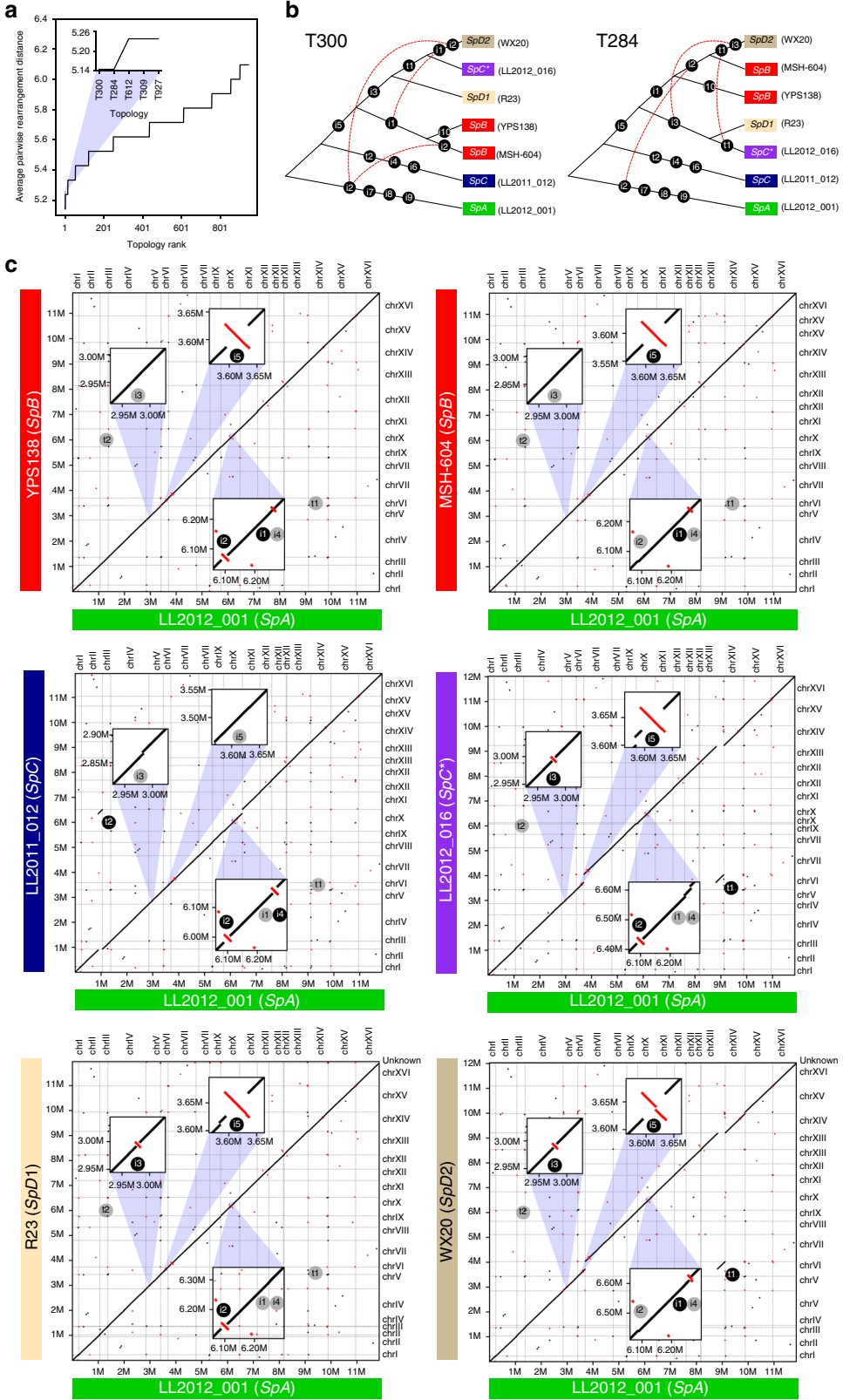

strains are early generation hybrids from crosses between *SpB* and *SpC\**. The early-generation hybrid hypothesis is further supported by the number of apparent crossing-overs per chromosome (average 10.2), which is in the reported range of 2–10 crossing-overs per meiosis for *S. cerevisiae*[21]. However, recombination rate was recently shown to be about 40% lower in *S. paradoxus*

compared to *S. cerevisiae*, which would push the origin of these strains a little further back in time[22]. These observations support a recent hybrid origin for the *SpD* strains, which have likely undergone only few rounds of meiosis.

In the *SpD* strain WX21, many regions could not be assigned to either *SpB* or *SpC\** ancestry because they appeared to be

**Fig. 2** Genome rearrangements support that *SpD* results from the backcross of the hybrid species *SpC*\* with its parent *SpB*. **a** The average pairwise distance (in number of rearrangements) between genomes for each of the 945 tree topologies tested. Topologies are ranked according to a maximum parsimony criterion, the best topologies involving the least rearrangements. The five best topologies (blue shaded area and inset) are highlighted. **b** Two bifurcating tree topologies (T300 and T284) best describe the evolutionary relationships among North American *S. paradoxus* according to the maximum parsimony criterion from **a**. Both show that *SpD* is more closely related to *SpC*\* and *SpB* than to *SpC*. Edges are labelled with the corresponding rearrangements, "iX" denoting inversions and "tX" denoting translocations. Red dotted lines connect conflicting rearrangements, i.e. rearrangements that occur on two or more branches. **c** Pairwise synteny across seven North American *S. paradoxus* genomes using *SpA* as a reference. Insets highlight examples of genomic rearrangements. Inversion i3 on chromosome V and translocation t1 (VItXIII) support the *SpC*\*-*SpD* close relationship; inversion i5 on chromosome VI supports the *SpB*-*SpC*\*-*SpD* relationship; inversion i1 on chromosome X supports the *SpB*-*SpD* relationship; inversion i4 on chromosome X is specific to *SpC*; and inversion i2 on chromosome X is an example of conflicting rearrangement, being shared by *SpA*, one *SpB* strain and one *SpD* strain. Red segments represent inversions. Alignments longer than 1 kb are shown

heterozygous, with haplotypes from both parental lineages (last outer ring, Fig. 3a). This is the only noticeable case of heterozygosity across all the North American strains reported so far. This suggests that although rare, outcrossing does occur in these lineages. Low levels of heterozygosity in other *SpD* strains are not in contradiction with the early hybridization hypothesis, since *Saccharomyces* haploids can readily autodiploidize through mating-type switching. Thus, spores from first-generation *SpD* hybrids could have become homozygous after a single meiosis. The breakpoints between heterozygous haplotypes indicate that WX21 likely originated from a cross between two strains of the *SpD2* sub-group (Supplementary Figure 12). This result shows that admixture is still ongoing in this group of strains and generates genomic diversity.

**Recent origin of the hybrid *SpD* group**. We estimated the divergence time of both hybridization events (*SpC*\*, *SpD*) by comparing the genetic divergence of *SpB*-like regions that are *SpD*-specific and *SpC*\*-specific relative to *SpB*. We found that the mean divergence of *SpD* from *SpB* is significantly lower than the mean divergence between *SpC*\* and *SpB* (Fig. 3c; *T1*, Mann-Whitney *U*-test, *P*-value < 2.3e−05). In addition, we used a Bayesian approach[23] to estimate the timing of the two hybridization events (Fig. 3d). Using the concatenated sequences of nine genes that were admixed from *SpB* into *SpC*\* and from *SpC*\* into *SpD* (Supplementary Tables 4-5), we estimated the divergence of *SpC*\* from *SpB* to be 18,500 years ago (±5,000). This is in the same range as the dating initially performed on different sets of loci[15]. The divergence of *SpD* from *SpB* was estimated to be of 2600 years (±750) (Supplementary Table 6). Further, we estimated that diversification within the *SpD* lineage could have occurred very recently, about 1800±750 years ago, with most of the strains not being older than a few hundred years. The young age of the *SpD* lineage estimated by these methods is consistent with its very limited geographic distribution (Fig. 1a).

**Impact of hybridization on molecular and growth phenotypes**. The persistence of a hybrid lineage and its development into a separate species depend on several factors, including hybrid-specific phenotypes and reproductive isolation from parental species[4,24]. The hybrid species *SpC*\* is phenotypically distinct and partially reproductively isolated from its two parental species *SpB* and *SpC*[15]. The very recent hybridization of *SpC*\* with *SpB* to form the *SpD* group allowed to investigate two important aspects of inter-species hybridization: the emergence of new phenotypes and reproductive isolation. By assuming that the current state of *SpD* is a good approximation for the early stage that follows hybridization, comparing *SpD* and *SpC*\* allows to examine the phenotypic evolution of a typical *S. paradoxus* hybrid lineage over two time frames separated by about 10,000 years of evolution.

To characterize the phenotypic profile of the *SpD* group relative to other lineages, we measured the growth of 229 strains, including 12 *SpD* strains, in 24 growth conditions that probe a diversity of potential metabolic performances. These conditions comprise various carbon sources, nitrogen sources, chemical compounds or incubation temperatures (Supplementary Data 3). We computed two measures of colony growth based on highly resolved growth curves: integrated growth through time (Area Under the Curve, AUC) and maximal growth rate (Maximum Slope, MS; Supplementary Figure 13). Hierarchical clustering of the strains for these two traits across conditions indicated that most *SpD* strains are phenotypically similar to each other and distinct from the other lineages (Fig. 4a). Linear discriminant analysis further suggested a linear combination of conditions that distinguishes *SpD* from the other lineages (Supplementary Figures 14–16).

Since outcrossing is rare in yeast[25], the long-term persistence of a new lineage likely depends on how it initially performs in the local environment as opposed to how reproductively isolated it is. We therefore compared the performance of the hybrid lineages *SpC*\* and *SpD* (respectively *SpD1* and *SpD2*) with their respective parental lineages across the 24 conditions (Fig. 4b). Despite their genetic similarity to *SpC*, *SpC*\* strains showed an overall stronger growth than *SpC* strains and weaker than *SpB* strains (Fig. 4c), as previously observed by Leducq et al.[15]. *SpC*\* outperformed both parental lineages when grown on mannose for the MS trait but not in any other condition. This suggests that *SpC*\* could show hybrid superiority in specific conditions, which may have contributed to its long-term success. The *SpD1* clade performed significantly worse than *SpB* and *SpC*\*, while *SpD2* was worse than *SpB* but not significantly different from *SpC*\* (Fig. 4c). Neither *SpD1* nor *SpD2* significantly outperformed both of their parental lineages in any condition tested (Supplementary Figure 17). *SpD* exhibited phenotypic novelty but has an apparent lack of overall advantage compared to its parental lineages. This suggests that *SpD* might be at a disadvantage across a wide range of growth conditions and thus may reflect the extrinsic reproductive isolation between its parental lineages. Since *SpD* is mostly allopatric to *SpC*\* and *SpC*, its lack of performance relative to the sympatric *SpB* may be the most consequential. We cannot exclude that the *SpD* group could be more competitive in conditions that we did not test as it is difficult to determine what the fitness determinants of each lineage are in their respective environments. Overall, these results clearly show that *SpD* hybrid strains have unique growth profiles and that hybridization has large consequences on complex phenotypes.

To further examine the role of hybridization in generating new phenotypes in the two hybrid lineages *SpC*\* and *SpD*, we performed gene expression profiling on a subset of 24 strains in two replicates each (Supplementary Tables 7-8, Supplementary Data 4, Supplementary Figures 18-19). The genome-wide gene expression profile of *SpD* showed an intermediate phenotype in

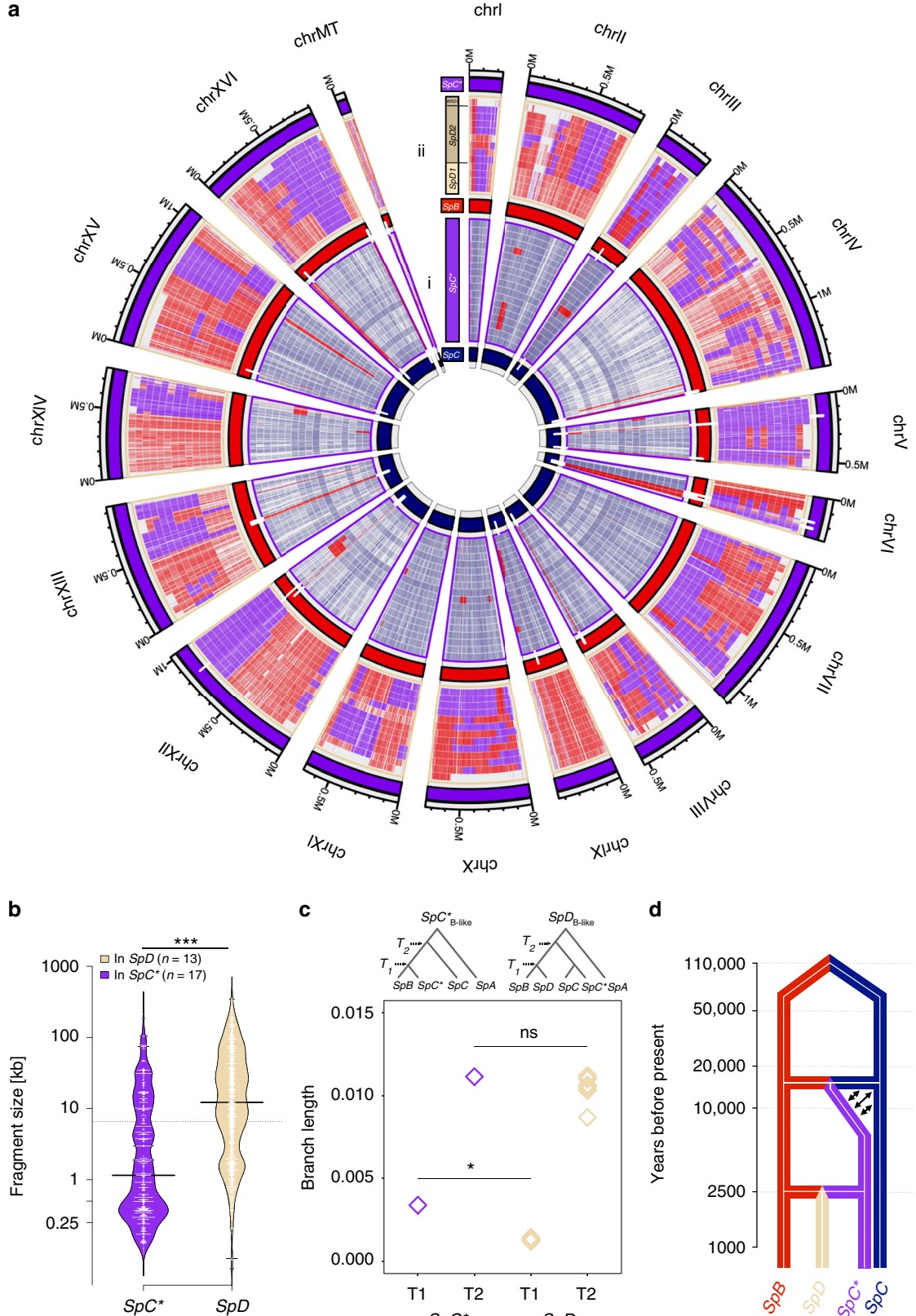

the first principal component, consistent with its almost 50% *SpB* and 50% *SpC\** genome composition (Fig. 4d, Supplementary Figures 20-21). Gene Ontology (GO) enrichment analysis from pairwise differential expression (Supplementary Figure 22) revealed that the *SpD* lineage is particularly distinguished from both *SpB* and *SpC\** in terms of functions related to amino acid

biosynthetic/catabolic processes, peptidyl-serine dephosphoryla-tion (*SpD1* to *SpB* and *SpC\**) and ribosome biogenesis (*SpD2* to *SpB* and *SpC\**) (Supplementary Figure 22; Supplementary Data 5). *SpC\** clustered strongly with its closest parent *SpC* in the two first components, as expected from their shared ancestry of ~97%. It appears that transcription profiles therefore largely reflect the

**Fig. 3** Genome-wide pattern of introgression in the young and old hybrids. **a** Shared ancestry in 17 $SpC^*$ and 13 $SpD$ strains. Genomic positions are colored according to parental genotypes. (i) 51 genes contain partially or completely introgressed genes from $SpB$ (white bar) and are fixed in $SpC^*$. (ii) Fragments of $SpB$ or $SpC^*$ ancestry in 13 $SpD$ strains. 12 $SpD$ strains show large ancestral fragments shared with $SpB$ or $SpC^*$ (ring 1–2; orientation: from inside to outside). Strain WX21 (ring 13) has many heterozygous regions across the genome, where the separation between $SpB$ and $SpC^*$-like ancestral was not possible (blank elements) (Supplementary Figure 12). Rendered by white bars (chrV, VI, and XII) are nine genes that $SpC^*$ inherited from $SpB$ and that are shared between all $SpD$ strains with $SpC^*$. **b** Length distribution of introgressions from $SpB$ shows shorter blocks on average in $SpC^*$ than in $SpD$, supporting a more recent origin of the latter (median of 1.1 kb in $SpC^*$ and 12.2 kb in $SpD$ respectively; Mann-Whitney $U$-test, $P$-value < 2.2e−16). The genome-wide proportion of regions with $SpB$-ancestry in $SpC^*$ is 3.1% on average, while it is 44.7% on average for $SpD$. **c** Relative time of origin of $SpD$ and $SpC^*$ ($T_1$) estimated from the divergence of $SpB$-like regions in each admixed lineage ($SpD$ or $SpC^*$) with corresponding regions in $SpB$. $T_2$—divergence time of $SpB$-like regions in admixed lineages with corresponding regions in $SpC$. Times were estimated for one strain of $SpC^*$ and all 13 strains of $SpD$ from the same sampling location. Diamonds depict mean estimates of divergence time across all contiguous $SpB$-like regions with more than 1000 informative sites. In all $SpD$ strains, $SpB$-like regions are significantly younger than in $SpC^*$ (mean $T_1$, Mann-Whitney test, $P$-value < 2.3E−05), but divergence time of $SpB$-like regions with $SpC$ remains the same in two admixed lineages (mean $T_2$, Mann-Whitney $U$-test, $P$-value > 0.23). **d** Dating the hybridization events. The lineage $SpD$ emerged only recently (~2500 years ago) from the crosses between $SpB$ and $SpC^*$. Arrows indicate initial backcrosses between $SpC^*$ and $SpC$

genomic composition of the admixed strains, more than the phenotypic profiles, potentially because most transcriptional changes result from neutral evolution and thus reflect genetic divergence[26] (Fig. 1). However, these transcriptional changes could also play a causal role in the poor performance of hybrid strains in specific conditions as those are associated with core metabolic (amino acids) and growth (ribosome biogenesis) processes.

**$SpD$ shows partial reproductive isolation with other lineages**. The persistence of $SpD$ as a genetically distinct group requires that it is reproductively isolated from its parental species. Liti et al.[27] observed a positive correlation between nucleotide divergence and reproductive isolation in *Saccharomyces sensu stricto* yeasts, showing that reproductive isolation accumulates with time. This is also the case in our study system[15,19]. However, crosses between the parental lineage $SpB$ or $SpC$ and the hybrid species $SpC^*$ resulted in similar degrees of spore survival (38 and 49% respectively) even though $SpC^*$ has higher sequence identity with $SpC$ (see also: Leducq et al.[15],Charron et al.[19]). Chromosomal rearrangements and genetic incompatibilities can accelerate the onset of reproductive isolation between lineages[28]. The isolation of $SpC$ and $SpC^*$ was indeed previously shown to result at least partly from chromosomal rearrangements, explaining the deviation from the general trend observed within the genus[27]. $SpD$ could also benefit from such rearrangements that cause partial isolation from its parents.

We thus sought to measure the degree of reproductive isolation of $SpD$ and observed high fertility among $SpD2$ strains (mean = 94%; $n = 3$; Fig. 4e). However, $SpD1s$ showed a decreased fertility when crossed with each other (mean = 65%, $n = 3$). The same degree in fertility was also observed after the direct sporulation of wild $SpD1$ and $SpD2$ homothallic isolates (Supplementary Data 6). Since $SpD1$ also exhibited weaker overall growth in the phenotypic screen, these strains may bear an excess of deleterious alleles or allele combinations, which could lower both spore viability and colony growth measured in various environmental conditions.

We found that $SpD1$ and $SpD2$ show relatively high fertility when crossed with the young hybrid species $SpC^*$ (Fig. 4e and Supplementary Data 6). Fertility dropped when these $SpD$ strains were crossed with more diverged lineages, such as $SpC$. Surprisingly, backcrosses of $SpD$ with the parental lineage $SpB$ also show very low spore survival ($SpD2$, mean = 28% (4–47%), $n = 6$; $SpD1$, mean = 38% (24–48%), $n = 6$), similar to what we observe in crosses between the older lineages $SpB$ and $SpC$. This partial reproductive isolation between $SpB$ and $SpD$ could enable the persistence of both lineages in sympatry on the long term.

One notable exception are crosses between $SpD$ strains and strains of a rare group (~1%) of $SpB$ strains, called $SpBf$, which harbor an important translocation between chromosomes VI and XIII (VItXIII). These crosses showed a spore survival (Supplementary Data 6) similar to what is observed for crosses with $SpC^*$. Previous data showed that $SpBf$ strains are the closest $SpB$ relatives to $SpC^*$, because they share the VItXIII translocation (t1, Fig. 2) and this translocation was shown to be correlated with spore inviability in crosses between $SpC^*$ and $SpC$[15]. Our results, however, show that the higher fertility of $SpD$-$SpBf$ crosses may not be due to the presence of the VItXIII translocation[15]. Indeed, we detected the VItXIII translocation in $SpD2$, $SpC^*$ and $SpBf$ strains but not in $SpD1$ (Supplementary Figure 23). Therefore, the presence of the translocation likely does not explain $SpBf$'s higher fertility with $SpD1$ than with $SpD2$. Other genomic rearrangements, detected or not detected in the structural analysis (Fig. 2c), could play important roles.

Overall, $SpD$ strains show highly reduced fertility with the two most abundant lineages in North America, $SpB$ and $SpC$. This applies particularly to crosses with $SpB$ strains, which occur in the region where $SpD$ is found. Despite its young origin, $SpD$ appears to have achieved levels of reproductive isolation with $SpB$ similar to those between $SpB$ and $SpC$[29,30], providing an opportunity to examine how hybridization may lead to rapid isolation. The sufficient reduction in fertility could therefore enable the persistence of the $SpD$ lineage and further its independent and divergent evolution. The relatively high fertility with its other parental species $SpC^*$ may compromise this independent evolution. However, since $SpC^*$ is extremely rare outside of the $SpB$-$SpC$ sympatric region in the St-Lawrence valley and given the low frequency of sexual reproduction in *S. paradoxus*, $SpD$-$SpC^*$ mating remains highly unlikely. Geographical barriers could therefore play the major role in this case.

## Discussion
In this study, we present a case of recurring hybridization in a eukaryotic microorganism. We show that a new group of diverse hybrid strains has evolved from a backcross between a young hybrid species and its parental lineage. Our findings highlight that recurring hybridization events in nature contribute to genomic, phenotypic and potentially species diversity.

The evolutionary conditions that make recurrent hybridization and the emergence of new lineages possible may be very restrictive and are not completely known. However, Blanckaert and Bank[30] recently showed that reproductive isolation could arise very rapidly through incompatibilities, where specific combinations of parental haplotypes in the $F_2$ generation cause reproductive isolation with the parental lineages. This could be accelerated in yeast through local inbreeding or selfing allowed by

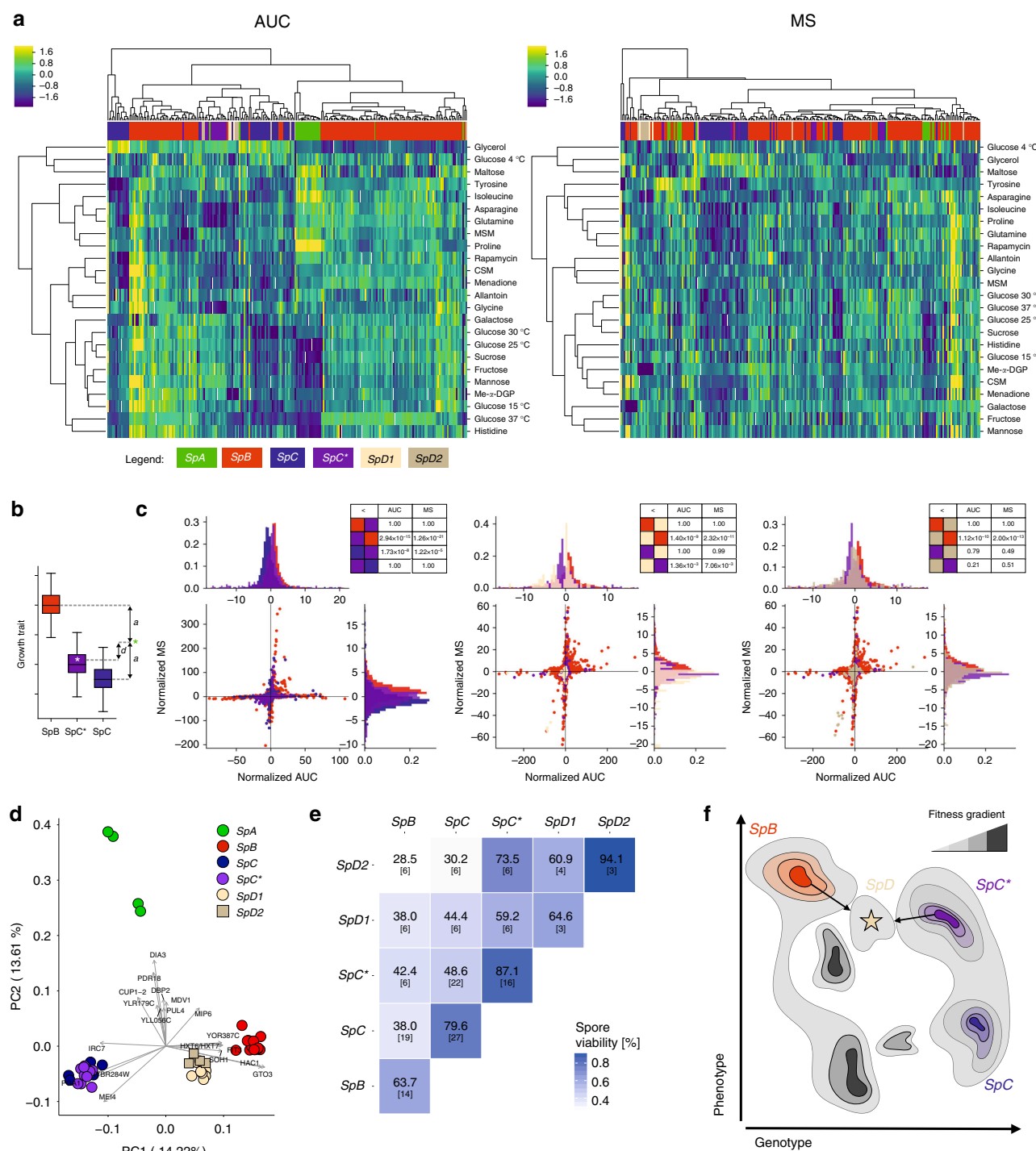

mating type switching within a few cell divisions[25], rapidly making incompatibility loci homozygous and thus eliminating the costly phase of segregation of incompatibilities. The nature of these incompatibilities remains to be identified and might involve mechanisms including elevated mutation rate[31], mitochondrial-nuclear interactions[32,33] and chromosomal rearrangements[15].

Rapid hybrid speciation within a few generations based on hybrid phenotypes was recently shown to occur by Lamichhaney et al.[34]. Our findings further support that this could be a fast process and reveal that repeated hybridization could occur within a time frame of a few generations of sexual reproductions. The SpD lineage displays novel growth and transcriptome phenotypes, along with partial reproductive isolation with other lineages.

These conditions could be sufficient for the emergence of another hybrid species. One challenging question is whether and how the young SpD group will persist through time given its apparent lack of growth advantage compared to the parental species. This observation contrasts with many studies showing that $F_1$ yeast hybrids usually show hybrid vigor across a broad range of conditions[35,36]. The overall poor performance of the young SpD hybrids relative to the expected hybrid vigor and compared to the older SpC*, could represent a transient state, in which gene combinations are being sorted by natural selection. Following a scenario proposed by Mallet[37], SpD could represent an early hybridization event with individuals that are distant from a phenotypic optimum, a stage of homoploid hybrid speciation that

**Fig. 4** Phenotypic divergence and reproductive isolation of *SpD*. **a** Hierarchical clustering of normalized colony growth profiles in 24 conditions based on AUC or MS. The heatmaps correspond to *z*-score normalization of the rows (conditions) excluding outliers. CSM: complete synthetic medium, Me-α-DGP: methyl-α-D-glucopyranoside, MSM: minimal synthetic medium. **b** Value normalization used to compare hybrids with their parental lineages across conditions. The difference *d* of a given strain's value (white star) to the mid-parent value (green star) is divided by the half-difference *a* between the parental distributions medians. **c** Distributions of normalized AUC and MS for the three hybrid lineages *SpC\**, *SpD1*, and *SpD2* and their parental lineages. *P*-values of one-sided Mann-Whitney *U*-tests are detailed in the tables. **d** PCA on genome-wide gene expression levels. The two first dimensions cumulatively explain 27.8% of the total variance. The 20 genes most contributing to these dimensions are shown. *SpC* and *SpC\** are grouping together, while *SpD* strains show an intermediate transcription profile between *SpC/SpC\** and *SpB*. **e** Reproductive isolation measured from spore viability tests of intra- and inter-lineage crosses from four lineages (*SpD* is separated into *SpD1* and *SpD2*). High spore viability was detected for all intra-lineage crosses. Inter-lineage crosses generally suffer from low viability. However, inter-lineage crosses between recently emerged or close related lineages (*SpC\**-*SpD*) indicate higher fertility than crosses between older lineages (*SpB*-*SpC*). The number of crosses in each case is shown in brackets. **f** Conceptual framework adapted from Mallet[37] to illustrate how hybrids may explore genotypic and phenotypic spaces and fitness peaks. In this framework, *SpD* may be a group of individuals that need to exploit further genotypic and phenotypic space to gain higher fitness and to potentially occupy novel ecological niches. The color gradient corresponds to variation in fitness with darker colors representing higher fitness. This shows several potential unoccupied and potential niches (dark gray) with fitness peaks that could lead, if reached, to the persistence for *SpD*

could be inevitable. These hopeful monsters could take advantage of new gene combinations in future generations that may eventually make reaching an optimum possible (Fig. 4f). Long term persistence would therefore depend on the sorting of genetic diversity currently found in this group of strains. Also, the spatial density of *Saccharomyces* strains has been shown to be very low[38] such that growth competition may be limited. Other factors than growth rate, such as cell survival to specific stresses or the exploitation of specific resources, may be greater determinants of long-term maintenance. It remains challenging to study the ecological and genomic parameters that contribute to microbial evolution in a natural setting. A key element for future investigations will therefore be the study of fitness components in natural conditions, for instance through the implementation of common garden-experiments[39] and the ability to follow the evolution of genotypes through time[16]. The use of the model system *S. paradoxus* may be one of the few systems to offer this opportunity.

## Methods

**Strain collection**. The 74 novel *S. paradoxus* strains sequenced in this study were sampled between 2014 and 2016 and collected from different substrates and locations (Supplementary Data 1-2). Samples were isolated and enriched for *Saccharomyces* strains with the method described in Sniegowski et al.[14] and confirmed to belong to *S. paradoxus* by sequencing a ~750 bp genomic region including the loci *ITS1* and *ITS2* (Oligo forward: GTTTCCGTAGGTGAACCTGC; Oligo reverse: ATATGCTTAAGTTCAGCGGGT)[40], following the protocol described in Leducq et al.[41].

**Whole-genome sequencing by Illumina**. Genomic DNA was extracted from independently grown colonies on solid rich media (YPD, Yeast extract, Peptone, Dextrose) with DNeasy Blood & Tissue Kit (Qiagen, Hiden, Germany). Libraries were prepared with the Illumina Nextera XT kit (Illumina, San Diego, USA) following the manufacture's protocol. Libraries and input DNA were quantified with AccuClear Ultra High Sensitivity dsDNA Quantitation Kit (Biotium, Fremont, USA) using the Fusion Optics (SPARK, TECAN, Männedorf, Switzerland). Pooled libraries were sequenced on a single lane of HiSeqX (150PE, Illumina, San Diego, USA) at the Genome Quebec Innovation center (Montréal, Canada). The 74 strains were sequenced with an average genome-wide coverage of ×45(Supplementary Figure 1, Supplementary Data 2). Raw sequences are accessible at NCBI (bio project ID PRJNA479851).

**Read mapping and variant calling**. Genome sequences from the 244 previously sequenced *S. paradoxus* strains from North America were downloaded and added to our newly sequencing strains[15,16] (NCBI accession: PRJNA277692, PRJNA324830; Supplementary Data 1). Independently sequenced libraries of the same strains from the Leducq et al.[15] study were combined. Raw reads were mapped on the reference genome of CBS432[18] using Bowtie2 v2.1.0[42] with default settings and the local alignment option. Strains 16_053B and 16_187A were removed because 16_053B showed poor sequencing results and 16_187A revealed low alignment rates (coverage < 1) due to a contamination. Sequences were sorted and indexed with Samtools v1.8[43].

Variants were flagged using default variant quality filters implemented in the Platypus variant caller v0.8.1[44]. The program Vcftools v0.1.15[45] was used to remove variants that showed either of these features: (1) low quality per depth ratio (QD filter); (2) in regions with low root-mean-square mapping quality (MQ filter); (3) had quality below 20 Phred scores (Q20 filter); (4) failed the strand bias filter (strandBias filter); (5) fell in the low-complexity regions (SC filter); (6) were supported by reads with bad quality bases or were present on one strand only (badReads filter); (7) fell in regions supported by too many haplotypes (HapScore filter) or (8) showed lower than expected frequencies (alleleBias filter). Additionally, genotypes with Phred scores below 20 were masked. INDEL variants were removed using a custom python2.7 script. Further, we used vcflib v1.0.0 with the *vcfallelicprimitives* function to split Multiple Nucleotide Polymorphisms (MNPs) into Single Nucleotide Polymorphisms (SNPs), and Bcftools v1.6[43] with the *norm* function to remove duplicated positions. Finally, only biallelic variants were retained with two alleles across all strains. The overall alignment rate to the European *SpA* reference genome CBS432[18] was 82.4 %, with an estimated average coverage of 32.6 over the 316 genomes (Supplementary Figure 1).

**Global phylogeny and principal component analysis**. A subset of 25,280 variants regularly spaced across the genome was selected from a total of 205,206 variants across the 316 *S. paradoxus* strains. A maximum likelihood phylogenetic tree was built with RAxML v8.2.9[46]. The tree was estimated with 1000 rapid bootstraps, under generalized time-reversible (GTR) model with gamma distribution of substitution rates. The program Adegenet v2.1.0[47] was used to perform a Principal Component Analysis (PCA) in which we calculated the median values of each principal component for each lineage and the 20th and 80th percentile to identify the PCs that split closely related lineages (e.g. *SpC* and *SpC\**; Supplementary Figure 2).

**Inference of population structure**. The population genetic structure in the *SpB* lineage was estimated using the Bayesian clustering method implemented in Structure v2.3.4[48] (Supplementary Figure 4). The filtered set of 25,280 SNPs was used and only sites that were variable in the *SpB* lineage were selected. Additionally, adjacent SNPs in strong linkage disequilibrium (LD) were removed using Plink v1.90b4.4[49]. Parameters—indep-pairwise 50 5 0.5 were used to remove one SNP from a pair in a 50 SNP window, with a step of 5 SNPs, if the estimate of LD (r2) for the pair was higher than 0.5. The list of the remaining SNPs in *vcf* format was converted into Structure format using PGDSpider v2.1.1.0[50]. Structure analysis was performed for several number of clusters, respectively ranging from $k = 2$–8, with 10 replicates per each cluster. Each replicate analysis was run for 30,000 steps with the first 10,000 steps excluded as a burn-in. *Delta K* statistic was calculated (Supplementary Figure 4B) to identify the most likely number of clusters with Harvester v.0.6.94[51]. Multiple replicate analyses of each cluster were then aligned using CLUMPP v1.1.2[52].

**Test of admixture scenarios**. In the PCA (Supplementary Figure 2), strains from the lineage *SpC\** group closely with *SpC*, whereas *SpD* is intermediate between *SpB* and *SpC/SpC\** for the first two PCs. To confirm that *SpD* is a hybrid lineage and to identify signatures of shared ancestry with other lineages, we tested 15 different models using genome-wide polymorphism data. The models assume admixed ancestry of *SpD* and/or admixed ancestry of *SpC\** or no admixture events (Supplementary Figure 5). Models were compared with the data by means of *f4* statistics calculated in the qpDstat program from AdmixTools v1.0.1 package[53]. Their fit to the data was evaluated using admixture graph fitting implemented in admixture-graph R package v1.0.2[54]. In admixturegraph, the models are represented as admixture graphs that generate expectations of *f4* statistics based on graph edge lengths and admixture proportions, and the parameters of the graphs are optimized using a cost function that minimizes differences between expected and observed

*f4* statistics. The fitted *f4* statistics for each of 15 models are shown in Supplementary Figure 6. Prior to *f4* calculations, only variants present in at least 4 strains in every lineage (*SpA*, *SpB*, *SpC*, *SpC\**, and *SpD*) were selected. Models were ranked according to the minimal error, which is an estimate of a cost function for the given model for the best fitting values of *f4* (Fig. 1e). When running value optimization 100 times, model M01 yielded fitted fitted values within 3 standard errors of observed statistics in 100% of runs, whereas model M02 yielded similar values only in 58% of runs. Other models never yielded a perfect fit with the data. To make sure that the two recognized groups of *SpD* (*SpD1* and *SpD2*) follow the same admixture history, we separately tested all 15 models using only *SpD* strains from the *SpD1* and *SpD2* group. We found the same models supporting admixture history of these groups (Supplementary Figures 7-9). Model M01 yielded fitted values within 3 standard errors of observed statistics in 100% of runs for *SpD1* and *SpD2*, whereas model M02 yielded similar values only in 86 and 65% of runs for *SpD1*, and *SpD2*, respectively. Other models never yielded a perfect fit with the data.

**Long-read sequencing**. Six strains from the North American lineages were sequenced with the Oxford Nanopore technology using two FLO-MIN107 flow cells (R9.5). The subset of strains passed on each run is shown in Supplementary Table 1. The 1D Native barcoding genomic DNA protocol with kits SQK-LSK108 and EXP-NBD103 was used. Genomic DNA was extracted with two successive phenol-chloroform extractions followed by ethanol precipitation. One microgram (run 1) or 1.5 µg (run 2) of genomic DNA per strain was fragmented using Covaris g-TUBEs at 6000 rpm (run 1) or 4200 rpm (run 2) in an Eppendorf 5424 (Hamburg, Germany) centrifuge. Fragmentation was assessed by agarose gel electrophoresis. No FFPE DNA repair step was performed. For run 1, all DNA recovered after adapter ligation was pooled and loaded on the flow cell due to low yield, while for run 2 the amounts of DNA were adjusted to achieve approximately equal representation of the six strains in the two combined libraries. Sequencing was performed using the default script provided with the MinKNOW software v1.7.14.

**De-novo assembly of long-read sequencing data**. The data from the two independent sequencing runs were basecalled, demultiplexed and trimmed separately, after which they were combined. Reads that passed the MinKNOW quality filter (*pass* directory) in fast5 format were basecalled and demultiplexed using Albacore v2.0.2 with the following parameters: −f FLO-MIN107 −k SQK-LSK108 −o fastq −−barcoding. Adapter trimming was performed using Porechop v0.2.2 (github.com/rrwick/Porechop) with parameter −b set for demultiplexing. Only the reads assigned to the expected barcode by both Albacore and Porechop were kept. De novo genome assembly was performed using SMARTdenovo with parameters −J 1000 −c 1.

The following polishing steps were performed on the resulting draft assemblies. First, the raw assemblies were polished using Nanopolish v0.9.0[55] with the trimmed Nanopore reads. The basecalled sequences of trimmed nanopore reads in fasta format were mapped on the raw assemblies using BWA MEM v0.7.16a-r1181[56] with parameter −x ont2d. The Nanopolish variants program was run with parameters −p 2 −min-candidate-frequency 0.1 to yield polished consensus assembly sequences. Second, Illumina reads for the same strains (PE 100 bp or 150 bp) were used to perform additional correction on the Nanopolish-processed assemblies using Pilon v1.22[57]. Adapter and barcode sequences for the Illumina TruSeq sequencing kit were retrieved and combined in a custom reference library (TruSeq_custom_retrieved.fa). Then Illumina reads were trimmed using Trimmomatic v0.36[58] with parameters PE −phred33 ILLUMINACLIP: TruSeq_custom_retrieved.fa:2:30:10 TRAILING:3 SLIDINGWINDOW: 4:15 MINLEN:36. The trimmed reads were mapped on the Nanopolish-processed assemblies using BWA MEM with default parameters. Optical and PCR duplicates were marked using Picard MarkDuplicates (broadinstitute.github.io/picard) with default parameters and the alignment files were filtered using Samtools view v1.5[43] with parameter -F set to 0 × 704, after which Pilon was run with the −diploid parameter. The corrected contigs were aligned with the PacBio assembly of a North American *S. paradoxus* strain (YPS138) produced by Yue et al.[18] using Mauve v2.4.0[59] with the progressiveMauve algorithm and default parameters. Contigs were manually assigned to chromosomes and were re-ordered using a custom Python v3.6.3 script to yield the final assemblies. Assembly sizes and N50 are presented in Supplementary Table 1.

**Evolutionary analysis of genomic rearrangements**. Genomic rearrangements were identified by performing a multiple genome alignment with rearrangements using Mauve. The progressiveMauve algorithm was used with default parameters to align the six SMARTdenovo assemblies along with the PacBio assembly of one *SpB* North American strain (YPS138; Kuehne et al.[60]) produced by Yue et al.[18]. Unless otherwise stated, all further analyses were performed using custom Python v3.6.3 scripts. Genomic blocks of local collinearity were extracted from the *.backbone* file produced by Mauve and filtered to exclude blocks smaller than 500 bp, or which were not conserved among the seven genomes. Block permutations were exported in the GRIMM[61] format, manually excluding the contig boundaries that corresponded to chromosome splits as those were unlikely to correspond to real splits. The comprehensive set of rooted bifurcating tree topologies with six labelled leaves was generated using the NetworkX package[62] in Python and

exported to Newick format. MGRA v2.2.1[63] was used to reconstruct ancestral genomic block permutations for each tree topology independently, allowing to map the number of rearrangements that occurred on each branch. The tree topologies implying the smallest total number of rearrangements were selected as the ones most likely describing the evolutionary relationships among the genomes.

**Detection of introgressed regions and genes**. Genome-wide fasta-alignments of the 316 *S. paradoxus* strains were prepared to identify introgressed regions. Samtools v1.8 and Bcftools v1.8[43] were used to calculate the genotype likelihood from the bam-formatted alignment files, to call variants and to create single fasta files for each individual strain. Here, the *S. paradoxus SpA* strain CBS432[18] was used as a reference genome. A custom Python script was used to combine the single fasta files into chromosome-separated alignment files. Nucleotide diversity within and divergence between lineages were calculated from genome-wide fasta files (model: JC69) in Pangorn v.2.3.1[64] using all 316 individuals.

The detection of introgressed fragments of *SpB*-origin in the *SpC\** strains was based on two approaches. First, for each gene, we calculated the genetic distance (model: JC69) of each *SpC\** strain to each *SpB* and *SpC* strain. We then determined if each *SpC\** strain was genetically closer to either *SpB* or *SpC* at that locus. We marked a gene as being fixed and introgressed from *SpB* in the *SpC\** lineage, when all *SpC\** strains had a smaller genetic distance to all *SpB* strains than to any *SpC* strain. As input data, we used the 17 *SpC\** strains (as identified by PCA, the global phylogeny and in Leducq et al.[15]) and compared them to the 213 *SpB* strains and 50 *SpC* strains from our current data set of 316 strains.

The program HybridCheck v1.0[65] was used to perform a Triplet-Test, where the strains UWOPS-79140 and LL2012_027 represented the lineages *SpB* and *SpC* to precisely delimit regions of shared ancestry in the 17 *SpC\** strains. The initial window size was set to 1 bp and was further extended in 1 bp steps along the chromosomes to identify introgressed blocks. Fixed introgressed regions were defined by identifying sequences that showed shared ancestry with *SpB* in all *SpC\** strains (Fig. 3a). The detection of introgression in the 13 *SpD* strains was done as described above, with the *SpB* strain UWOPS-79140 and the *SpC\** strain LL2012_016 to distinguish regions of different ancestry. This data was plotted using the R library ggBio[66].

The average coverage and frequency of heterozygous sites within 1 kb windows along the genome was calculated for strains that showed large regions of missing information in HybridCheck (Supplementary Figure 12). This analysis was computed in Samtools v.1.8[43] with the command *depth* using sorted *bam*-files. Heterozygous sites where filtered from the variant file (*vcf* format), which was used in the first place to construct the global phylogeny.

Gene ontology enrichment for the introgressed genes was performed in GOrilla[67] using the complete list of genes present in our *S. paradoxus* strains as the reference list (Supplementary Figure 11, Supplementary Table 3).

**Fixed introgressed genes of *SpB*-origin in *SpC\****. Previous analysis of Hénault et al.[17] using the data from Leducq et al.[15], identified 105 introgressed genes in H0 regions (= fixed and introgressed regions of *SpB*-origin) of *SpC\**. The number of introgressed genes in this study however is lower than the number detected by Hénault et al.[17] (Supplementary Table 2). The difference is likely linked to the two distinct detection methods for introgressions and the different sets of strains (11 strains in Leducq et al.[15] vs. 17 strains in this study). The detection method in Leducq et al.[15] was based on a 5 kb window approach (non-sliding). This method likely increased the number of potentially introgressed genes by including all complete or partial, not introgressed genes found within 5 kb windows showing overall signature of introgression. The sliding window-based approach in HybridCheck was based on a 1 bp step size with an initial window size of 1 bp and therefore, identified shorter introgressed fragments with higher confidence, which led to a precise detection of introgressed genes in *SpC\** strains.

**Distribution of introgressed fragments in *SpC\** and *SpD***. The size of fragments that showed *SpB* ancestry along the genomes of the 17 *SpC\** and 13 *SpD* strains was calculated using the data from HybridCheck. Fragments of shared ancestry that were within 5 kb distance were concatenated (Fig. 3b).

**Dating of the origin of *SpD* lineage**. To determine which of the two admixed lineages *SpD* or *SpC\** is more recent, the approximate age of the hybrid lineage *SpC\** and *SpD* was assessed from pairwise nucleotide divergence (Fig. 3c). This was done by estimating nucleotide divergence of *SpB*-like regions between *SpD*/*SpC\** and *SpB* ($T_1$). To make sure that the differences are not caused by mutation rate variation between lineages, divergence time of the same *SpB*-like regions between *SpD*/*SpC\** and *SpC* was estimated ($T_2$). To date the emergence of *SpD*, *SpB*-like regions present in *SpD* but absent in *SpC\** were extracted. $T_1$ was calculated using sequence alignments of 5 lineages with topology (((*SpB*, *SpD*),(*SpC\**,*SpC*)),*SpA*), where *SpA* is an outgroup. $T_1$ corresponds to split between *SpB* and *SpD*, and $T_2$ corresponds to split between (*SpB*, *SpD*) and (*SpC\**, *SpC*). *T* was estimated by counting private derived variants accumulated on tree branches of 5 lineage

 

topologies in the following way:

$$T_1 = \frac{1}{N} \times \frac{BAAAA + ABAAA}{2} \qquad (1)$$

$$T_2 = \frac{1}{N} \times \frac{\frac{BAAAA+ABAAA}{2} + BBAAA + \frac{AABAA+AAABA}{2} + AABBA}{2} \qquad (2)$$

where $N$ is the number of sites in the alignment, A are the ancestral and B are the derived variants. To date the emergence of $SpC^*$, $SpB$-like regions present in $SpC^*$ were extracted. For sequence alignments of 4 lineages with the topology $((SpB, SpC^*),SpC),SpA)$, an analogous estimation of $T_1$ and $T_2$ was performed, where $T_1$ corresponds to split between $SpB$ and $SpC^*$, and $T_2$ corresponds to split between $(SpB,SpC^*)$ and $SpC$. $SpB$-like regions in $SpD$ and $SpC^*$ were determined using HybridCheck, with $SpB$ and $SpC$ strains as parental strains (see: analysis on the detection of introgression). Only regions having at least 1000 informative sites in the alignment were considered. After excluding positions with missing information, we obtained ancestral and derived variant patterns for each region with the fasta2dfoil program in the dfoil package[68]. Times were estimated for all available 13 $SpD$ strains, $SpC^*$ strain from the same location (16_199Ci), and one strain per $SpB$, $SpC$, and $SpA$.

Further, the Bayesian method implemented in Beast 2.0[23] was used to estimate the divergence time of $SpD$ from other lineages (Fig. 3d). Two independent dataset were prepared: (1) Nine genes of $SpB$ ancestry in $SpC^*$ (fixed in $SpC^*$) and in $SpD$ (fixed in all $SpD$; $SpC^*$ ancestry) were concatenated (Fig. 3a, Supplementary Table 4). (2) In addition, a second dataset comprised of concatenated sequence of 4 fragments from chromosomes V (position: 50–130 kb), XII (position: 860–1000 kb), XIV (position: 510–650 kb) and XVI (position: 510–780 kb) for a total size of 630 kb was selected. The fragments of the second dataset (2) were identified in HybridCheck as being inherited from $SpC$ to $SpC^*$ and subsequently from $SpC^*$ to all the $SpD$ strains. For both datasets, the following analysis was limited to 66 strains that were randomly chosen as representatives of all 5 lineages, but included all $SpC^*$ and $SpD$ genomes (Supplementary Table 5). In Beast, a substitution rate of 1.67e−10 was chosen from estimates in Zhu et al.[69] for S. cerevisiae and previous studies that estimated the divergence time for $SpC^*$ in Leducq et al.[15]. A fixed clock that assumed equal substitution rates among branches was used with a calibration time of 100,000 years ago (±10,000 years) for the split of the lineages $SpB$ and $SpC$, which corresponds to the onset of the last glaciation (~100,000 years ago). The analysis was performed using $15 \times 10^6$ MCMC iterations with 30% of burn-in and statistical sampling every 10,000 iteration (Supplementary Table 6). We used FigTree v1.4.2[70] to visualize the trees.

**High-throughput yeast colony growth.** A total of 229 sequenced North American S. paradoxus strains available in our collection were arranged into four random arrays of 1536 positions on OmniTray plates (Thermo Fisher Scientific, Waltham, USA) of YPD solid media (1% yeast extract, 2% tryptone, 2% glucose, 2% agar) using a BM5-BC-48 colony processing robot (S&P Robotics Inc., Toronto, Canada). Each strain was represented in 12 replicates randomly distributed across the four arrays. Strains 16_236B, 16_277B and UWOPS-80-13 were excluded because their colony texture did not allow proper handling by the robotic pin tool, and strain 16_187A was excluded after the identification of a contamination in the sequencing data. The two outer rows and columns of the arrays were filled with wild S. cerevisiae colonies to avoid plate border effects on growth. The 1536 arrays were replicated on 25 different media (Supplementary Data 3) and incubated at 25 °C (unless otherwise stated): complex media (1% yeast extract, 2% tryptone, 2% agar) with 2% fructose, glucose, galactose, maltose, mannose, methyl α-D-gluco-pyranoside, sucrose or 3% glycerol; complex media with 2% glucose incubated at 4 °C, 15 °C, 30 °C or 37 °C; synthetic media (2% glucose, 0.174% yeast nitrogen base, 2% agar) with 0.5% asparagine, glycine, glutamine, histidine, isoleucine, lysine, proline, tyrosine or ammonium sulfate (AS); synthetic sucrose allantoin medium (2% sucrose, 0.174% yeast nitrogen base, 2% agar, 0.005% allantoin); synthetic complete media (2% glucose, 0.174% yeast nitrogen base, 2% agar, 0.5% AS, 0.134% complete drop-out) alone or supplemented with 50 μM menadione or 200 ng/μl rapamycin. After the colonies reached replicable size (48–96 h, depending on the condition), they were replicated on the same media. For the media incubated at 25 °C, plates were placed into an automated incubator-imager (S&P Robotics Inc., Toronto, Canada) and photographed at 2 h intervals. For the other temperatures, the plates were incubated in separate incubators and taken out briefly to be photographed on the colony-processing robot, first at 2 h intervals (6 h for plates at 4 °C) and at increasing time intervals afterwards.

**Parameters imputation and dataset filtering.** Plate images were processed using the gitter R package v1.1.1[71] using a standard plate image as value for the argument ref.image.file and setting plate.format to 1536. All the downstream analyses were performed using custom Python v3.6.3 scripts. Time values assigned to each picture were corrected to attribute a single time value to all pictures taken during a same imaging round. Colony entries that had a size of 0 or flags for abnormal circularity or overlapping boundaries were removed. The filtered dataset contained between one and 12 replicates for each of the 315 333 (strain × condition × imaging time point) combinations, with an average of 11.74 replicates per combination and

0.012% of the combinations having less than three replicates. The median of the 12 or less replicate colony sizes for any given strain was transformed in $\log_2$ and normalized by subtracting the initial median $\log_2$ colony size to yield growth curves. For complex media with fructose, glucose, galactose, mannose, methyl α-D-glucopyranoside and sucrose at 25 °C, growth curves were truncated at 80 h because colonies on these media had achieved stationary phase. Each growth curve contained between 184 and 852 individual data points, corresponding to between 29.72 and 100% of the total expected data points (between 73.66 and 100% when excluding synthetic medium with histidine). Three metrics were extracted from these curves: area under the curve (AUC), maximum slope (MS) and endpoint colony size (ECS). AUC was computed by fitting a cubic spline to the raw data points using the UnivariateSpline function from the scipy Python package v1.1.0 and by integrating the resulting function using the spline's integral method. The MS was computed as the 98th percentile of the set of linear regression slopes fitted in 5-timepoints wide overlapping sliding windows with a correlation coefficient $r > 0.8$. ECS was the colony size at the last time point of each growth curve. Only AUC and MS were considered due to the high correlation between AUC and ECS (Pearson's $r = 0.98$). Although AUC and MS yield overall similar patterns in multiple factors analysis (MFA), both were considered separately for further analysis due to their lower correlation (Pearson's $r = 0.86$). Growth on synthetic medium with lysine was extremely slow for all strains, thus this condition was excluded from the analysis. Strain MSH-1S11 was excluded due to visible contamination of the colonies. Additionally, strains UCD 62-186, UCD 62-268 and YPS695 were excluded because their AUC, MS or ECS value was 0 in at least one condition. A preliminary PCA analysis on the AUC data identified 8 strains that grouped closely on PC1 (43% of total variance explained): 14_164C, 14_169C, 14_177C, 15_005C, 16_033B, LL2012_004, R23 and yHKS267. These strains had very poor growth on all the conditions tested and were thus excluded from the analysis.

**Colony growth data analysis by dimensionality reduction.** Several dimensionality reduction methods were applied on AUC and MS data. PCA was performed using the decomposition.PCA function from the Python package scikit-learn v0.19.1 with argument svd_solver = 'full'. LDA was performed using the discriminant_analysis.LinearDiscriminantAnalysis function from scikit-learn with argument solver = 'eigen'. Additionally, the general agreement between AUC and MS data was assessed by applying multiple factor analysis (MFA), using AUC and MS values as groups of variables. MFA was performed using the MFA function from the Python package Prince v0.4.0 with arguments rescale_with_mean = False, rescale_with_std = False, n_components = 48, n_iter = 10, copy = True, engine = 'auto', random_state = 42.

**Hierarchical clustering of growth values.** Strains were clustered according to their AUC or MS values across conditions. Hierarchical clustering was performed using the function cluster.hierarchy.linkage from scipy, with complete linkage as the method and Spearman correlation as the distance metric. Heatmaps were generated using the clustermap function from the Python package Seaborn v0.8.1[72], with values normalized by z-score transformation within conditions. Heatmaps were plotted by setting clustermap's argument robust = true to restrict color mapping to the values falling between the 2nd and 98th percentiles of the data.

**Growth comparison between $SpC^*$, $SpD$, and parental lineages.** AUC and MS values for the hybrid lineages $SpC^*$ and $SpD$ (respectively $SpD$ sub-group 1 ($SpD1$) and $SpD$ subgroup 2 ($SpD2$)) were compared to that of their respective parental populations. To include all conditions in a global comparison, we normalized the growth values of all relevant strains in a given condition to center them around the intermediate point between parental distributions and to scale them according to the amplitude of the difference between the parental distirbutions (Fig. 4b). For each strain (hybrid or parental), we computed the scaled difference $d$ between its value and the midpoint between the median values of the two parental lineages for each condition tested:

$$d = \frac{x_s - \frac{1}{2}[median(X_{P1}) + median(X_{P2})]}{\frac{1}{2}|median(X_{P1}) - median(X_{P2})|} \qquad (3)$$

where $x_s$ is the value of a given strain (hybrid or parental), and $X_{P1}$, $X_{P2}$ are the distributions of values for the first and second parental lineages, respectively. The differences between distributions of $d$ values for the lineages $SpC^*$ and $SpD$ and parental populations was tested using one-sided Mann-Whitney U-tests.

**Strain material and cell cultures for RNAseq analysis.** A subset of 24 diploid strains that were representatives of all five S. paradoxus lineages were used for gene expression profiling (Supplementary Table 7, Supplementary Data 4). Two biological replicates per strain were grown at 25 °C in synthetic media supplemented with allantoin (0.174 % Yeast Nitrogen Base, 2% Sucrose, 0.005% allantoin). Allantoin was used because it was shown to best represent variation in growth on Maple sap, one of the natural environment in which S. paradoxus is found in North American deciduous forests[73]. Two hundred and fifty milliliter cultures at 0.03 initial $OD_{600/ml}$ were prepared from pre-cultures in exponential phase and cultures

 

grew until reaching OD$_{600/ml}$ of 0.6–0.7, were centrifuged at 4 °C to collect cell pellets and were immediately frozen in liquid nitrogen (stored at −80 °C).

Cell pellets were re-suspended in 2.5 ml lysis buffer (10 mM Tris-HCL pH 7.4, 100 mM NaCl, 30 mM MgCl$_2$) and cell droplets were prepared in liquid nitrogen. Frozen droplets were grinded in a RETSCH Mixer MM400 (Retsch, Haan, Germany) for 2 min and the chambers were place in liquid nitrogen to keep the cell extracts frozen. This procedure was repeated 15 times. Afterwards, samples were centrifuged for 10 min (13,000 g) at 4 °C. The supernatants (lysate) were transferred to new tubes, frozen in liquid nitrogen and stored at −80 °C until further processing.

**RNA extraction, RNA library preparation and sequencing**. Two hundred microliter of lysate for each sample was purified using the RNA Clean & Concentrator-25 kit (R1018; Zymo Research, Irvine, USA) following the manufacturer's instructions. DNA contamination was eliminated by performing the DNase I In-Column treatment (E1010; Zymo Research). Samples were eluted in 25 μl DNase/RNase-free water. The Quantseq 3′ mRNA kit (Lexogen, Vienna, Austria) was used for library preparation[74]. Fourty eight libraries were sequenced on 4 Ion Torrent chips with an expected yield of >60 million reads per chip. The final average coverage over the 6000 genes was more than 800 reads per gene. Adapters, sequences of less than 30 bp in length, Poly-A tails and sequences with quality score lower than 15 were removed using the program Cutadapt[75]. Read quality statistics were retrieved from the program FastQC[76]. Raw sequences can be downloaded under the NCBI bio project ID PRJNA480398.

**Genomes de-novo assembly and annotation**. Six reference genomes were prepared (Supplementary Figure 18), respectively one for the lineages *SpA* (LL2012_026A), *SpB* (UWOPS-79140), *SpC* (LL2012_027), *SpC** (LL2012_016) and two for the lineage *SpD* (SpD1 = R24, SpD2 = WX19), as the latter was shown to cluster in two distinct groups[17]. *De-novo* assemblies were performed on the raw sequencing reads from Leducq et al.[15] and Xia et al.[16]. Megahit v1.1.2[77] was used to assemble reads into large contigs with the k-mer option set to 41, 51, 71, 91, 111, 131, 159, 187, and 215, which led to 706–1323 contigs, with a mean N50 of 137,674 (122,691 to 154,133). Contigs were assembled into chromosomes using Chromosomer v.0.1.4[78] using the CBS432[18] sequence as reference. The program AUGUSTUS v. 3.2.3[79] predicted between 5398 and 5438 genes for the six reference genomes. A custom Python script was used to identify orthologous genes between the six reference genomes. Briefly, this script combined a syntenic approach with the detection of orthologs based on blastp[80] results from all query proteins against all reference proteins. The position of a given gene and the sequence similarity to ORFs of the reference was used to identify orthologous genes in the six reference genomes. Gene prediction was further extended to non-annotated ORFs detected from transcription data and increased the annotations of ORFs to 5315-5355 genes per reference genome (Supplementary Table 8).

**Read counts and differential expression**. The program BWA v.0.7.17[81] was used to align the 48 transcriptomes to the six reference genomes of the same lineage (lineage-specific, Supplementary Data 4). Since the library preparation and subsequent sequencing produced only one fragment per gene that was located downstream in the 3′ UTR region, a custom script counted the number of reads per gene. Briefly, a first window of 400 bp (100 bp in 3' end plus 300 bp of 3′ UTR region for each gene) was generated. In this first window, the shape of the distribution of reads was detected with the aim to identify the highest peak representing the expression of a gene. The window was further extended in 100 bp steps in both directions until the distribution was rendered by a valley of low read counts (<10 reads or <10% of reads from the highest detected peak in the extended window).

The analysis of differential expression was performed in R[82] using the DeSeq[83] pipeline (Supplementary Figures 19-22). In DeSeq, raw read counts were normalized using the rlog function, which transformed the raw read count data into log$_2$ scale and corrected for variation coming from different library sizes. We performed PCA on the genes that were common to all individuals of the six lineages (in DeSeq; Fig. 4d, Supplementary Figures 20-21). GO-enrichment analyses of pairwise expression comparison (Supplementary Figure 22, Supplementary Data 5) were performed using Gorilla[67].

**Heterothallic strain construction and spore viability**. The mating type switching locus (HO) of five different *SpD* strains, respectively M2 and R22 of *SpD1*, and WX20, WX21, R21 of *SpD2*[16] was deleted using the procedure and oligonucleotides described in Charron et al.[19]. Hygromycin (HPH), Nourseothricin (NAT) and Kanamycin (KAN) deletion cassettes were amplified with primers CLOP40-E9 (ACATCCTTATAGGCAGCAATCAATTCCATCTAAACTTTAACCAGCTGAA GCTTCGTACGC; forward) and CLOP48-C9 (TTTATTACATACAACTTTTT TTTAATAATATACATATTGCATAGGCCACTAGTGGATCTG; reverse) or CLOP40-E8 (TTAATTACATAACAATTTTTTTTTATAATATACATATTGCA TAGGCCACTAGTGGATCTG; reverse). After transformation, successful deletions of HO were verified by PCR using primers CLOP48-C11 (ACAGAAG CTTGTTGAAGCGC; forward) and HPH_B (GTCGCGGTGAGTTCAGGCTT; reverse - HPH cassette) or KAN_B_R (CTGCAGCGAGGAGCCGTAAT; reverse -

KAN cassette) or NAT_B (CGGTAAGCCGTGTCGTCAAG; reverse - NAT cassette). Diploids were sporulated and segregants were tested for mating type using primers verifMATa_F (ACTCCACTTCAAGTAAGAGTTTG; forward - MATa) or verifMATalpha_F (GCACGGAATATGGGACTACTTCG; forward - MATα) and verifMATa/alpha_R (AGTCACATCAAGATCGTTTATGG; reverse).

Crosses among the five *SpD* strains and with 11 additional strains from other lineages (Supplementary Data 6) were performed following the protocol in Charron et al.[19]. Spore viability of crosses was measured from the proportion of visible colonies from 24 tetrads per cross. Spores that form visible colonies after a 72-h incubation period at 30 °C were considered viable. In addition, fertility of wild, diploid *SpD* strains was quantified (without any genetic modification) by sporulation and subsequent dissection (Supplementary Data 6, Fig. 4e).

**Detection of the chromosomal translocation VItXIII in *SpD***. To detect the presence of the translocation between chromosome VI and the right arm of chromosome XIII (VItXIII)[15] in *SpD* strains, the scaffold TA04_6134 from Leducq et al.[15] was used. This scaffold was generated from sequencing data from the *SpC** strain 2012_018 and has a length of ~172 kb. It harbors a translocation from chromosome XIII that fused to the right arm of chromosome VI in the *SpC**. This translocation was shown to be unique to all *SpC** individuals and present in a few *SpB* individuals denoted as *SpB-fusion* (SpBf). To assess the presence of this translocation in the *SpD* strains, a subset of 42 strains from all five lineages was generated (including the three *SpBf*, all 17 *SpC** and the 13 *SpD* strains) and raw sequencing reads were mapped to the scaffold TA04_6134 using Bowtie2 v2.1.0[42]. The coverage per single position was calculated with *depth* in Samtools v1.8[43] and averaged in 1 kb windows along the scaffold. The presence of reads at the junction (~20 kb) that connect the two fused sequences was used as evidence for the existence of the translocation (Supplementary Figure 23). From the alignment files (*bam*), a variant (*vcf*) file with the same parameters as above was prepared using Platypus v0.8.1[44] and vcftools v0.1.15[45]. Phylogenetic relationships where assessed from variants in the scaffold region of 36 kb to 57 kb in R[82] with the packages Ape[84] and Phangorn[64].

**Code availability**. All custom code will be available from the corresponding authors upon request.

**Reporting summary**. Further information on experimental design is available in the Nature Research Reporting Summary linked to this article.

## Data availability
The data that support the findings of this study are available from the corresponding authors upon request. All sequencing data generated in this study are available in the NCBI BioProject and Sequence Read Archive (SRA) databases under the accession numbers PRJNA479851 (short reads for 74 genomes), PRJNA514804 (long reads for 6 genomes) and PRJNA480398 (48 transcriptomes). Sequences from previous studies can be found under the accession numbers PRJNA277692 (Leducq et al.[15]) and PRJNA324830 (Xia et al.[16]).

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

## Acknowledgements

This work was supported by a fellowship from the Quebec Network for Research on Protein Function, Engineering, and Applications (PROTEO) to C.E. and by the Natural Sciences and Engineering Research Council of Canada (*NSERC*) Discovery Grants to C.R.L., L.M.K., and J.B.A.; M.H and G.C were supported by graduate fellowships from NSERC and A.F. by Genome Canada and Genome Québec (bioSAFE). C.R.L. holds the Canada Research Chair in Evolutionary Cell and Systems Biology. We thank Yasir Ahmed, Ian Caldas, Andrew Clark, Sofie Delbare, Anne-Marie Dion-Côté, Jean-Baptiste Leducq, and Sarah Lower for their contribution to sampling. We thank the sequencing platform at IBIS for technical support and Nadia Aubin-Horth, Johan Hallin, and other members of the LandryLab for comments.

## Author contributions

C.E. and C.R.L. planned this study. C.E., A.F., M.H., and M.B. performed the population genomics analysis. C.E. and M.B. performed the transcriptomic analysis. M.H. performed the phenotypic screen and the long-read sequencing analysis. G.C. performed the reproductive isolation experiments. C.E. wrote the paper with contributions from M.H., A.F., G.C., M.B., L.M.K., J.B.A. and C.R.L.

## Additional information

**Competing interests:** The authors declare no competing interests.

