## [Peer Review File · Nature Communications]

Reviewers' Comments:

Reviewer #1:

Remarks to the Author:

This is an interesting and comprehensive analysis of a newly discovered population of *Saccharomyces paradoxus*, which shows signs of being an incipient species. The new population (SpD) was formed by hybridization between the predominant North American population (SpB) and SpC*, which itself is a hybrid and is present at a low frequency in the area (near Toronto) where SpD was found. The authors report a very careful analysis whose main conclusion is that the hybrids SpD and SpC* were formed by two separate hybridization events, about 10,000 years apart. Their results indicate that yeasts located in natural habitats that host multiple closely related species are experiencing repeated cycles of hybridization and genome resolution. Phenotypic analysis shows that the new population SpD does not appear to have a growth advantage over SpB which predominates in its locale, but SpD is partly reproductively isolated from SpB. One of the most intriguing aspects of this study is that multiple chromosome rearrangements are segregating in these yeast populations, which may be contributing to reproductive barriers. Overall, this is a highly detailed analysis, using state-of-the-art genomics and population techniques, that characterizes this novel yeast species in exquisite detail. My only comments concern some aspects of the data presentation that were unclear.

I had a lot of difficulty understanding Figure 3A. Its legend is not adequate. It took me a while to figure out that part "i" shows 17 SpC strains, and part "ii" shows 13 SpD strains, so it would be helpful to write this information beside the "i" and "ii" labels. I still can't find the "introgressed genes from SpB rendered in black" (if they're introgressions from SpB, shouldn't they be colored red in the rings in part i? But I can't see red in most of them, only on chr XIII and XVI... are they red but outlined in black edges that are too thick to allow the red to be seen?). Line 166 says that the two SpD subgroups (SpD1 and SpD2) are visible in Figure 3Aii... I can more-or-less see two patterns, but again it would be helpful to indicate which of the rings are SpD1 and which are SpD2.

Line 158: Does SpD here refer to both SpD1 and SpD2?

Figure 1B: Two clusters of SpD strains are marked in gray. Do these correspond to SpD1 and SpD2?

I had difficulty understanding Figure S23A. Why does it show two cartoons of chromosome VI (labeled "296 kb" and "320 kb"). The chromosomal regions involved in the rearrangement doesn't seem to be drawn to scale (the chr. XIII region is about half the chromosome size, i.e. ~400 kb, in the upper part, but less than 170 kb in the lower part). And if the junction region in the yellow box is only 15 kb (legend), then why is the bar containing it labeled 170 kb? What do the dashed lines and arrowhead mean? Where is the 110 kb fragment mentioned in the legend?

Reviewer #2:

Remarks to the Author:

Hybridization events between species have been shown to be a powerful way to evolve and lead to a broad genomic and phenotypic diversity. In this manuscript, the authors focus on the recurrence of hybridization events using yeast and more precisely *Saccharomyces paradoxus* as a model. By comparing the genome of a large collection of *S. paradoxus* isolates ($n=316$), they explored and defined the different hybridization events, which happened over time. They also looked at the impact of hybridization events on molecular and growth phenotypes. And finally, they studied the presence of reproductive isolation between the different defined subpopulations. Interestingly, they have shown that speciation in yeast may result from cycles of repeated hybridization events.

This is an interesting work leading to a more exhaustive view of key aspects of adaptation at different levels. The question is clearly stated and analyses provide clear results. This study is important because it provides solid experimental data to support insight into one of the key processes in evolution.

Overall, this is very nice story with interesting conclusions.

However, I have several points that need to be addressed:

1. line 78 - 92

the last paragraph of the introduction summarizes the previous studies, which lay the foundation of the presented work. However, this part is not well-written and very confusing without reading and knowing the previous articles. This part needs to be re-written to clearly replace the present study in its context.

2. line 99 - 100

"We sampled 203 more yeast isolates from Ontario..., including 38 *S. paradoxus*"

Does it mean that 165 isolates correspond to other species? The authors might want to give more details on that. Otherwise, this information seems to be irrelevant here.

3. line 101 - 102

"We assessed the population structure and genetic relationship of 316 strains..."

The numbers are a little bit confusing here. The authors included 38 additional *S. paradoxus* isolates but 72 fully sequenced genomes in the framework of this study. However, they looked at a total of 316 genomes. Again, this is very confusing.

4. line 187 - 189

"The early-generation hybrid hypothesis is further supported by the number of apparent crossing-overs per chromosome (average 10.2), which is in the reported range of 2-10 crossing-overs per 189 meioses for *S. cerevisiae*".

A recent study focusing on recombination in *S. paradoxus* found that the recombination rate is 40% lower in *S. paradoxus* compared to *S. cerevisiae* (Liu et al. MBE 2018). This part might be discussed in the light of this new result.

5. line 289 - 358

the part on reproductive isolation is too wordy and it needs to go directly to the facts. In addition, there is a main lack in this part, which is only partially discussed and considered: the structural variants (SVs). The authors detected some main SVs in specific strains. However, they also need to consider that some others, not detected, also might have an impact on the spore viability.

1 Response to reviewers:

2
3 **Reviewer #1 (Remarks to the Author):**

4
5 This is an interesting and comprehensive analysis of a newly discovered population
6 of *Saccharomyces paradoxus*, which shows signs of being an incipient species. The
7 new population (SpD) was formed by hybridization between the predominant North
8 American population (SpB) and SpC*, which itself is a hybrid and is present at a low
9 frequency in the area (near Toronto) where SpD was found. The authors report a
10 very careful analysis whose main conclusion is that the hybrids SpD and SpC* were
11 formed by two separate hybridization events, about 10,000 years apart. Their results
12 indicate that yeasts located in natural habitats that host multiple closely related
13 species are experiencing repeated cycles of hybridization and genome resolution.
14 Phenotypic analysis shows that the new population SpD does not appear to have a
15 growth advantage over SpB which predominates in its locale, but SpD is partly
16 reproductively isolated from SpB. One of the most intriguing aspects of this study is
17 that multiple chromosome rearrangements are segregating in these yeast
18 populations, which may be contributing to reproductive barriers. Overall, this is a
19 highly detailed analysis, using state-of-the-art genomics and population techniques,
20 that characterizes this novel yeast species in exquisite detail. My only comments
21 concern some aspects of the data presentation that were unclear.

22
23 **RESPONSE:**

24 We thank the reviewer for his/her very positive comments.

25
26
27 I had a lot of difficulty understanding Figure 3A. Its legend is not adequate. It took me
28 a while to figure out that part “i” shows 17 SpC strains, and part “ii” shows 13 SpD
29 strains, so it would be helpful to write this information beside the “i” and “ii” labels. I
30 still can’t find the “introgressed genes from SpB rendered in black” (if they’re
31 introgressions from SpB, shouldn’t they be colored red in the rings in part i? But I
32 can’t see red in most of them, only on chr XIII and XVI... are they red but outlined in
33 black edges that are too thick to allow the red to be seen?). Line 166 says that the
34 two SpD subgroups (SpD1 and SpD2) are visible in Figure 3Aii... I can more-or-less
35 see two patterns, but again it would be helpful to indicate which of the rings are SpD1
36 and which are SpD2.

37
38 **RESPONSE:**

39 The figure 3A has been modified according to the reviewer’s suggestions. We made
40 the figure more transparent and added additional information to identify the *SpC** and
41 the *SpD* strains. Further, we marked the *SpD1* and *SpD2* individuals in the figure.
42 The black colour, to show the introgressed and fixed genes, was replaced by another
43 colour (white) that shows the regions harbouring fixed genes. We also specified
44 which of the *SpD* strains is the heterozygous strain (*WX21*; last ring).

45
46
47 Line 158: Does SpD here refer to both SpD1 and SpD2?

48
49 **RESPONSE:**

50 We added additional information to specify that we meant *SpD1* and *SpD2* here.

51

52
53
54
55
56
57
58
59
60
61
62
63
64
65
66
67
68
69
70
71
72
73
74
75
76
77
78
79
80
81
82
83
84
85
86
87
88
89
90
91
92
93
94
95
96
97
98
99
100
101
102

Figure 1B: Two clusters of SpD strains are marked in gray. Do these correspond to SpD1 and SpD2?

RESPONSE:

Figure 1B was replaced by a similar figure, which allows the reader an easier overview about the relation and diversity of the different lineages. Additionally, we discriminate the sub-clades *SpD1* and *SpD2* in this phylogeny.

I had difficulty understanding Figure S23A. Why does it show two cartoons of chromosome VI (labeled “296 kb” and “320 kb”). The chromosomal regions involved in the rearrangement doesn’t seem to be drawn to scale (the chr. XIII region is about half the chromosome size, i.e. ~400 kb, in the upper part, but less than 170 kb in the lower part). And if the junction region in the yellow box is only 15 kb (legend), then why is the bar containing it labeled 170 kb? What do the dashed lines and arrowhead mean? Where is the 110 kb fragment mentioned in the legend?

RESPONSE:

The scales for the *cartoon*-chromosomes have been changed. 23A was also simplified (new representation of the translocation) to make it easier to understand. The bar of the heatmap was named according to its scaffold (as used in Leducq et al. 2016, Nature Microbiology, who initially identified this translocation in *SpC**). The text in the figure legend was modified accordingly.

Reviewer #2 (Remarks to the Author):

Hybridization events between species have been shown to be a powerful way to evolve and lead to a broad genomic and phenotypic diversity. In this manuscript, the authors focus on the recurrence of hybridization events using yeast and more precisely *Saccharomyces paradoxus* as a model. By comparing the genome of a large collection of *S. paradoxus* isolates (n=316), they explored and defined the different hybridization events, which happened over time. They also looked at the impact of hybridization events on molecular and growth phenotypes. And finally, they studied the presence of reproductive isolation between the different defined subpopulations. Interestingly, they have shown that speciation in yeast may results from cycles of repeated hybridization events.

This is an interesting work leading to a more exhaustive view of key aspects of adaptation at different levels. The question is clearly stated and analyses provide clear results. This study is important because it provides solid experimental data to support insight into one of the key processes in evolution.

Overall, this is very nice story with interesting conclusions.

RESPONSE:

We thank the reviewer for his/her very positive comments.

However, I have several points that need to be addressed:

103
104
105
106
107
108
109
110
111
112
113
114
115
116
117
118
119
120
121
122
123
124
125
126
127
128
129
130
131
132
133
134
135
136
137
138
139
140
141
142
143
144
145
146
147
148
149
150
151
152
153

1. line 78 - 92

the last paragraph of the introduction summarizes the previous studies, which lay the foundation of the presented work. However, this part is not well-written and very confusing without reading and knowing the previous articles. This part needs to be re-written to clearly replace the present study in its context.

RESPONSE:

We changed the wording and re-wrote the last paragraph to make it clearer for the reader. It now reads as:

“A recent population genomics study of *Saccharomyces paradoxus*, a budding yeast found worldwide on the bark of deciduous trees and their associated soils¹⁸, showed that a novel North American species evolved through hybridization about 10,000 years ago¹⁹. This hybrid species (*SpC**) originated from the secondary contact between the two most abundant species, *SpB* that occupies a large fraction of the continent, and *SpC*^{19,20}, which is found almost exclusively so far in the north east. *SpC** shows a unique profile of growth phenotypes¹⁹, occurs mostly in the zone of sympatry between its two parental species and shows reproductive isolation with both of them, which is caused at least partially by genome rearrangements. These findings revealed that hybridization occurred at least once between two incipient species (*SpB* and *SpC*) that originated a little more than 100,000 years ago and that it led to the formation of *SpC**. A recent study by Xia, et al.²¹ identified a novel group of strains, *SpD* (originally defined as “Clade d” and then mistakenly assigned to the *SpC** group), which exhibit signatures of genomic admixture, potentially involving the same parental species as *SpC**. Analyses by Hénault, et al.²² suggested that *SpD* could have arisen from a second hybridization between *SpB* and *SpC*, indicating that hybridization could have occurred multiple times in different locations^{21,22}.”

2. line 99 - 100

“We sampled 203 more yeast isolates from Ontario..., including 38 *S. paradoxus*”
Does it mean that 165 isolates correspond to other species? The authors might want to give more details on that. Otherwise, this information seems to be irrelevant here.

RESPONSE:

Since this information is not relevant, we removed this sentence.

3. line 101 - 102

“We assessed the population structure and genetic relationship of 316 strains...”
The numbers are a little bit confusing here. The authors included 38 additional *S. paradoxus* isolates but 72 fully sequenced genomes in the framework of this study. However, they looked at a total of 316 genomes. Again, this is very confusing.

RESPONSE:

This comment relates to the same concern as above. The sentences have been changed to be clearer.

“We assessed the population structure and genetic relationship from fully sequenced genomes of 316 *S. paradoxus* strains, which included 38 newly sampled strains (2016), 34 strains previously sampled, 91 genomes from Xia, et al.²¹ and 153

154 genomes from Leducq, et al. ¹⁹ (Figure 1, Supplementary Figure 1, Supplementary
155 Data 1-2)”

156

157

158 4. line 187 - 189

159 “The early-generation hybrid hypothesis is further supported by the number of
160 apparent crossing-overs per chromosome (average 10.2), which is in the reported
161 range of 2-10 crossing-overs per 189 meiosis for *S. cerevisiae*”.

162 A recent study focusing on recombination in *S. paradoxus* found that the
163 recombination rate is 40% lower in *S. paradoxus* compared to *S. cerevisiae* (Liu et al.
164 MBE 2018). This part might be discussed in the light of this new result.

165

166 RESPONSE:

167 We integrated (and cited) the findings of Liu et al. MBE 2018 in our results. This
168 section now reads as:

169 “The early-generation hybrid hypothesis is further supported by the number of
170 apparent crossing-overs per chromosome (average 10.2), which is in the reported
171 range of 2-10 crossing-overs per meiosis for *S. cerevisiae*^{27,28}. However,
172 recombination rate was recently shown to be about 40% lower in *S. paradoxus*
173 compared to *S. cerevisiae*, which would push the origin of these strains a little further
174 back in time²⁹. These observations support a recent hybrid origin for the *SpD* strains,
175 which have likely undergone only few rounds of meiosis.”

176

177

178 5. line 289 - 358

179 the part on reproductive isolation is too wordy and it needs to go directly to the facts.
180 In addition, there is a main lack in this part, which is only partially discussed and
181 considered: the structural variants (SVs). The authors detected some main SVs in
182 specific strains. However, they also need to consider that some others, not detected,
183 also might have an impact on the spore viability.

184

185 RESPONSE:

186 We made minor changes to the paragraph about reproductive isolation and also
187 integrated information, as suggested, about potential undetected SVs that can
188 contribute to spore viability. This section now reads as:

189

190 “The persistence of *SpD* as a genetically distinct group requires that it is
191 reproductively isolated from its parental species. Liti, et al. ³⁸ observed a positive
192 correlation between nucleotide divergence and reproductive isolation in
193 *Saccharomyces sensu stricto* yeasts, showing that reproductive isolation
194 accumulates with time. This is also the case in our study system^{21,26}. However,
195 crosses between the parental lineage *SpB* or *SpC* and the hybrid species *SpC**
196 resulted in similar degrees of spore survival (38% and 49% respectively) even though
197 *SpC** has higher sequence identity with *SpC* (see also: Leducq, et al. ²¹, Charron, et
198 al. ²⁶). Chromosomal rearrangements and genetic incompatibilities can accelerate the
199 onset of reproductive isolation between lineages³⁹. The isolation between *SpC* and
200 *SpC** was previously suggested to result at least partly from chromosomal
201 rearrangements, explaining the deviation from the general trend observed within the
202 genus³⁸. *SpD* could also benefit from such rearrangements that cause partial
203 isolation from its parents.

204

205 We thus sought to measure the degree of reproductive isolation of *SpD* and
206 observed high fertility among *SpD2* strains (mean=94%; n=3; Figure 4E). However,
207 *SpD1* showed a decreased fertility when crossed with each other (mean=65%, n=3).
208 The same degree in fertility we also observed after the direct sporulation of wild
209 *SpD1* and *SpD2* homothallic isolates (Supplementary Data 6). Since *SpD1* also
210 exhibited weaker overall growth in the phenotypic screen, these strains may bear an
211 excess of deleterious alleles or allele combinations, which could lower both spore
212 viability and colony growth measured in various environmental conditions.

213

214 We found that *SpD1* and *SpD2* show relatively high fertility when crossed with the
215 young hybrid species *SpC** (Figure 4E and Supplementary Data 6). Fertility dropped
216 when these *SpD* strains were crossed with more diverged lineages, such as *SpC*.
217 Surprisingly, backcrosses of *SpD* with the parental lineage *SpB* also show very low
218 spore survival (*SpD2*, mean=28% (4 to 47%), n=6; *SpD1*, mean=38% (24 to 48%),
219 n=6), similar to what we observe in crosses between the older lineages *SpB* and
220 *SpC*. This partial reproductive isolation between *SpB* and *SpD* could enable the
221 persistence of both lineages in sympatry on the long term.

222

223 One notable exception are crosses between *SpD* strains and strains of a rare group
224 (~1%) of *SpB* strains, called *SpBf*, which harbor an important translocation between
225 chromosomes VI and XIII (VItXIII). These crosses showed a spore survival
226 (Supplementary Data 6) similar to what is observed for crosses with *SpC**. Previous
227 data showed that *SpBf* strains are the closest *SpB* relatives to *SpC** because they
228 share the VItXIII translocation (t1, Figure 2) and this translocation was shown to be
229 correlated with spore inviability in crosses between *SpC** and *SpC*²¹. Our results
230 however show that the higher fertility of *SpD-SpBf* crosses is not due to the presence
231 of the VItXIII translocation²¹. Indeed, we detected the VItXIII translocation in *SpD2*,
232 *SpC** and *SpBf* strains but not in *SpD1* (Supplementary figure 23). Therefore, the
233 presence of the translocation likely does not explain *SpBf*'s higher fertility with *SpD1*
234 than with *SpD2*. Other genomic rearrangements, detected or not detected in the
235 structural analysis (Figure 2C), could play important roles.”

236